# Modelling and Optimisation of Multi-Stage Flash Distillation and Reverse Osmosis for Desalination of Saline Process Wastewater Sources

**DOI:** 10.3390/membranes10100265

**Published:** 2020-09-28

**Authors:** Andras Jozsef Toth

**Affiliations:** 1Environmental and Process Engineering Research Group, Department of Chemical and Environmental Process Engineering, Budapest University of Technology and Economics, Műegyetem rkp. 3, H-1111 Budapest, Hungary; andrasjozseftoth@edu.bme.hu; Tel.: +36-1-463-1490; 2Institute of Chemistry, University of Miskolc, Egyetemváros C/1 108, H-3515 Miskolc, Hungary

**Keywords:** modelling, multi-stage flash distillation, reverse osmosis, desalination

## Abstract

Nowadays, there is increasing interest in advanced simulation methods for desalination. The two most common desalination methods are multi-stage flash distillation (MSF) and reverse osmosis (RO). Numerous research works have been published on these separations, however their simulation appears to be difficult due to their complexity, therefore continuous improvement is required. The RO, in particular, is difficult to model, because the liquids to be separated also depend specifically on the membrane material. The aim of this study is to model steady-state desalination opportunities of saline process wastewater in flowsheet environment. Commercial flowsheet simulator programs were investigated: ChemCAD for thermal desalination and WAVE program for membrane separation. The calculation of the developed MSF model was verified based on industrial data. It can be stated that both simulators are capable of reducing saline content from 4.5 V/V% to 0.05 V/V%. The simulation results are in accordance with the expectations: MSF has higher yield, but reverse osmosis is simpler process with lower energy demand. The main additional value of the research lies in the comparison of desalination modelling in widely commercially available computer programs. Furthermore, complex functions are established between the optimized operating parameters of multi-stage flash distillation allowing to review trends in flash steps for complete desalination plants.

## 1. Introduction

In the year 2016, the total capacity of desalination plants was as much as 88.6 million m^3^/day on a global scale, providing daily supply of fresh water to about 300 million people [1]. For the most part, desalination processes are based on reverse osmosis (RO), multi-stage flash distillation (MSF), and multiple effect distillation (MED) [2,3,4]. Specifically, RO is present in 65% of the technologies within the total installed capacity worldwide. Thermal desalination operations account for 28% of the total capacity, 21% of which is MSF process and 7% is MED process [5]. The remaining capacity consists of nanofiltration, electrodialysis and other operations [5]. The distribution is also shown in Figure 1.

Although MED is thermodynamically more efficient than MSF, it is a less common method. MED units typically have a capacity of 600 to 91,000 m^3^/day, but involve high investment costs and energy consumption. The energy requirements of the MED process are 2–3 times higher than the energy needs of the RO process. One of the largest MED plants is Al Jubail in Saudi Arabia, with a total capacity of 800,000 m^3^/day (27 units with a capacity of 30,000 m^3^/day per unit) [5].

The typical energy requirements, investment costs, and cost of drinking water produced in relation to the 3 main desalination technologies described above are shown in Table 1 [6,7]. The energy demand of saline water RO (SWRO) is about 3–4 times less than that of MSF and 2–3 times less than that of MED, since models based on evaporation require not only electricity but also thermal energy, while RO uses only electrical energy to operate [8,9].

The historical turning point in the history of desalination was the appearance of multi-stage flash desalination (MSF), in Kuwait, dating back to 1957 [10]. The MSF system comprises three major parts: Heat Input Section, intermediate Heat Recovery Stages, and Heat Rejection Stage(s) where waste heat is discharged into the environment [11]. Figure 2 shows the flowsheet of the MSF method.

Saline water blended with recycled brine, preheated within the middle stages, is brought to working temperature (usually between 90–110 °C) with steam, in the Heat Input Section. At this point, it is transferred to the first stage, which is kept at a lower pressure than the heated saline water equilibrium pressure. Consequently, the saline water becomes partially vaporized. Steam generation removes energy from the mass of saline water, therefore cooling it to a certain extent. Furthermore, steam extraction results in higher salinity and higher boiling point in the case of the remaining saline water. The steam is transmitted to demisters, where transported droplets are eliminated, then it flows into a heat exchanger at the top of the stage. Consequently, the heat of the steam is passed over to the incoming saline water, whereby the steam cools down and condenses. By this process, the heat can be recovered with high efficiency.

Following condensation, water vapour is accumulated in the form of product water. The remaining saline water from the first stage is passed on to the subsequent step. In the second stage, there should be a lower pressure compared to that of the first stage, in order to facilitate further steam generation, as it compensates for lower temperature and higher salinity. The process is carried on, up to the last stage. The remaining saline solution is eventually cooled in the Heat Rejection Stage, thereby allowing it to be drained into the ocean. The incoming saline water is utilized for cooling in this final stage [11,12,13,14].

Although MSF is viewed as an advanced technology, further progress and improvements are underway. The long tubular distiller, which is equipped with an axially positioned saline flow system, may enhance unit capacity, reduce pressure drop, and increase overall performance, in contrast with the latest cross-flow solutions [15].

It has been stated by Rosso [16] that the performance of the plant is affected by the temperature of the heating steam and the temperature of the saline water, which can be described by the plant performance ratio (PR) or gained output ratio (GOR) as it can be seen in Equation (1) [17]. For MSF plants, a typical PR value is about 8 [18]:(1)$$PR=\frac{Distillate\;product\;\left[kg/h\right]}{Consumed\;steam\;\left[kg/h\right]}\left[-\right]$$

The production capacity of some currently operating RO water desalination plants and the price of produced drinking water are summarized in Table 2 [19]:

The opposite process, RO is one of the most important membrane technology operations. The membrane ideal for RO should correspond to the following requirements:relatively low cost,long-lasting and reliable structure,resistance to creep deformation,chemical and thermal stability in saline water,resistance to all kinds of fouling (inorganic, organic, colloidal and microbiological),resistance to oxidizing agents, especially chlorine,resistance to high temperature,high permeability to water,high salt rejection [10].

The original RO membranes were made from cellular acetate material. Since then, RO plants apply a variety of blends or derivatives of polyamides and cellular acetate. Plate, tubular, spiral-wound and hollow fiber membranes are the most popular RO modules [10,20]. Figure 3 shows the general structure of an RO plant.

The mentioned desalination processes, MSF and RO, were mathematically modelled and their transport phenomenon was described by several authors [9,21,22,23]. Ettouney and El-Dessouky [22] developed a computer package for the design and simulation of thermal desalination processes, MSF and evaporation. gPROMS software is used for effective design of the RO based desalination process, considering a wide range of salinity and seawater temperature, by Sassi and Mujtaba [24]. Skiborowski et al. [25] used also mixed-integer non-linear programming (MINLP) problem to rigorously optimize MED and RO. Lv et al. [26] investigated MSF with numerical simulation, CFD method and the flash chamber for multi-stage flash seawater desalination was optimized. Filippini et al. [27] introduced a mathematical model of hybrid RO and MED process. Detailed mathematical model of MSF was described by Rosso [16], Rao [28] and Helal et al. [29].

There were several attempts to run desalination processes in commercial software programs. SEEPUP package with FORTRAN addition was investigated for calculation of MSF by Husain et al. [30]. Aspen PLUS software was already used to simulate MED [31] and MSF [10] as well. Altaee [32] applied a computational model for estimating RO system design and performance: commercial software ROSA (Reverse Osmosis System Analysis) was investigated [32].

Not all simulator programs have built MSF calculation into their toolbars, which is also true for one of the most common chemical process simulators: ChemCAD does not have comprehensive desalination reference for complete plants. RO modelling is a membrane-specific process that requires complex computation, using calculations which are always valid for a given type of membrane.

The aim of this research is to investigate the desalination of saline process wastewater with MSF in ChemCAD professional flowsheet simulator, and compare the results with those of RO separation in WAVE design software from DuPont Company. Yields, operational properties, energetic factors, and complex functions between operating parameters were the basis for examination and comparison, which is considered as novelty in the case of mentioned computer programs.

## 2. Materials and Methods

The target of this work is to evaluate desalination methods of saline process wastewater (PWW). The initial concentration of the sample is motivated by an industrial separation problem: process wastewater from fine chemical industry with 4.5 V/V% or 45,000 ppm NaCl content. The desalination requirement is 0.05 V/V% or 500 ppm in both cases. 1000 kg/ feed stream was investigated.

For the evaluation of desalination processes, Yield [33] and Brine reduction are defined with the following equations:(2)$$Yield=\frac{Product\;\left[m^{3}/h\right]}{Feed\;\left[m^{3}/h\right]}\cdot 100\;\left[\%\right]$$
(3)$$Brine\;reduction=\left(1-\frac{Product_{NaCl}\left[V/V\%\right]}{Feed_{NaCl}\left[V/V\%\right]}\right)\left[-\right]$$

If the product emission limit was met, during the optimization of the process, the subsequent goal was to maximize yield and minimize energy.

### 2.1. Multi-Stage Flash Distillation

The MSF method is similar to multicomponent distillation, but there is no exchange of material between the counter-current flows. Actually, the MSF method is a flash evaporation method in vacuum, where the vacuum changes from one stage to the next and the evaporation temperature decreases from the first to the last stage [10]. MSF can be considered a non-linear recycling process with a closed-loop information flow in the simulator environment. ChemCAD is an advanced commercial software that allows the user to build and run steady-state and dynamic simulation models of chemical processes. MSF was rigorously steady-state modelled in flowsheet environment and optimized with dynamic programming optimization method [34,35,36] in ChemCAD 7.1.5 program environment.

The development of the MSF model in simulator environment involves the following steps:(1)Defining the flowsheet configuration by specifying:(a)Unit operations and(b)Process streams flowing between unit operations.(2)Specifying chemical compositions to be separated.(3)Choosing the thermodynamic model to represent the physical properties of the components and mixture in the method.(4)Specifying flow rates and thermodynamic conditions of the feed streams: i.e., pressure, temperature and phase conditions.(5)Optimizing operating conditions of unit operations in order to reduce NaCl content under 500 ppm of outlet water.

The thermodynamic model applied for the saline water is based on the NRTL equation but takes into account interactions of the electrolytic type effects. Thermodynamic calculations in the gas phase were carried out applying the Soave-Redlich-Kwong (SRK) equation [31]. The developed MSF model was verified with industrial reference before calculation. The operating parameters of AZ-ZOUR South MSF desalination plant from the work of Al-Shayji [10] were examined.

Figure 4 shows the structure of ChemCAD flowsheet of the MSF plant. As it is seen, the method can be divided into three part. The Heat Input (or Briner Heater) Section consists of a heat exchanger and loop modules. The Loop module regulated the order of feed flows [37]: at first steam was flown and then PWW was pumped into the MSF plant. This part consists of two input streams and two output streams. The first input flow is the ‘Brine recovery’ stream, which is used to heat to the required temperature. The second output flow is the steam with the conditions of 101.5 °C and 1.09 bar.

The Heat Recovery Section consists of interconnected liquid-liquid-vapour flash units (LLVFs), heat exchangers and mixer modules. The vapour inlet of LLVFs was condensed by heat exchanger modules. After the tenth recovery step, the plant follows with the heat rejection section, which consist of three stages. There are two addition flows: Make-up stream input and brine recovery output in the last LLVF stage. It must be mentioned, PWW is the cooling water in the case of the heat rejection section. The brine mass after the 13rd stage and the make-up stream were mixed together to give Brine recovery stream. It was recycled through the tubes of the heat recovery section. As a results of this procedure, desalted water can become available, which stream is called ‘water output’ in the flowsheet (see Figure 4). Figure 5 shows the optimization process of MSF simulation. At first, the initial data must be determined. This is followed by an extremely important part. Make-up and brine recovery flow rates and brine temperatures of stages have to be initialized. After that, the limit value can be reached using the method of ‘stage to stage calculation’. Flow rate and temperature are calculated in each stage and a sufficient number of recovery and rejection stages is determined to achieve an appropriate desalination effect.

### 2.2. Reverse Osmosis

Water Application Value Engine (WAVE 1.77a software was used for the calculation of RO. WAVE is a modelling software program that integrates three of the leading technologies (ultrafiltration, reverse osmosis and ion exchange resin) into one comprehensive platform [38]. The calculation method of WAVE is not public. However, Altaee [32] presented in detail the calculation of its Reverse Osmosis (RO) System Analysis module with 95% accuracy, which is based on experimental and practical contexts. The equations are discussed in detail in the paper of Altaee [32]. Thus, program calculations can be considered verified. With the method, salt retention, yield, permeate concentration, and flow rate can be estimated per membrane module. The calculation is based on Van’t Hoff equation, average concentration factor, and it also takes into account the pressure drop of the concentrate side, salt diffusion coefficient and concentration polarization [32].

The main steps of the calculation method are the following:(1)Estimation of feed pressure based on the feed osmotic pressure of the initial solution and the desired recovery rate of the system.(2)From the estimated feed pressure, estimation of the initial flow rate.(3)Calculation of initial recovery rate, permeate concentration and rejection rate for the module.(4)Next, estimation of the average salt concentration and water permeability to calculate an approximate flow rate for the membrane module.(5)The feeding of the concentrate from the first module to the second module.(6)In accordance with the previous steps, calculation of the flow rate and recovery for the second module, and then proceeding from module to module.

Figure 6 shows the flowsheets of RO modules in WAVE simulator. As it can be seen, the recycling of the concentrate stream was also investigated.

The main properties of applied RO membranes can be found in Table 3.

Figure 7 summarizes the calculation process of RO. The feed temperature was changed between 10 and 25 °C. The permeate pressure was taken to 10, 20 and 30 bars. The recovery represents the amount of permeate in the program.

## 3. Results and Discussion

### 3.1. Multi-Stage Flash Distillation

Table 4 shows the reference calculations of ChemCAD simulator. It can be seen that there is proper match between the results of industrial plant and ChemCAD simulations. Further parameters were also compared, which can be found in the Appendix A. Appendix A shows the temperature of recirculating brine entering each LLVF stage. Distillate produced from each stage in Ton/min can be found in Appendix A. Finally, Appendix A shows the outlet pressure from each stage. There is minor difference between industrial and simulated data, in all cases. Thus, it can be established that the developed model is verified and capable of the calculation of MSF.

Following the procedure of Figure 6, optimization of the MSF method was achieved. Seven different parameters were investigated in the function of flash stages:(1)Water output NaCl [V/V%](2)Water output temperature [°C](3)Water output pressure [bar](4)Water output flow rate [m^3^/h](5)Consumed steam [m^3^/h](6)Performance ratio: PR [–](7)Thermal energy [kWh/m^3^]

Figure 8 shows a two dimensional representation of tendencies. Figure 9 depicts complex functions between the mentioned parameters in three dimensional visualization, supplemented by the equation describing the plane.

It can be observed that the limit can be reached at the 13rd flash stage, therefore the optimum point can be found here. The 11 kWh/m^3^ value in the case of the 13-step-plant fits into the literature tendency, as it can be seen in Table 1. NaCl composition, temperature, pressure, flow rate and PR of Water output are almost on the same declining trend. The performance ratio was above 8 at the optimum location. In contrast, consumed steam and required Thermal energy are following the opposite trend, which represent the cumulative heat effect of the system. These tendencies are in line with experience in the literature [18,30]. The course of temperature and pressure are interrelated. In a binary system, one can be counted from the other. Furthermore, the values on the diagrams in Figure 8 can be traced back to the NaCl-water phase diagram.

Table 5 introduces the optimized results of the MSF method. The steam flow was 15 m^3^/h. The brine reduction was 0.99 and 11.7% can be reached in yield value. Figure 10 shows the pressure and temperature values of flash inputs and distillate products of the stages between the 1st and the 13rd. It can be seen that the curves are flattening close to the optimum region.

### 3.2. Reverse Osmosis

Table 6, Table 7 and Table 8 show the optimized results of SW30XHR-440i membrane. Table 9, Table 10 and Table 11 introduce the simulated data of SW30HRLE-440i membrane, and the results of SW30XLE-440i type RO membrane can be seen in Table 12, Table 13 and Table 14.

It can be affirmed that all three membranes met the limit value. As high as 0.99 brine reduction value can be reached in every case, which corresponds to 0.05 V/V% NaCl in permeate product. Studying the tables, low yield values are conspicuous. It means a characteristic result, which is the main disadvantage of RO compared to MSF. There is accordance between literature and simulated permeate flow rates or yields. SW30XHR-440i membrane has the lowest literature permeate flow rates and lowest simulated yields too. This is also the case for several other membranes, as it can be seen in Table 3.

The highest yield was obtained at the highest permeate pressure (30 bar) and 25 °C. In the case of SW30XHR-440i membrane yield of 8.6% can be achieved. 10.6% was the maximum yield with SW30HRLE-440i membrane and 12.6% was in the case of SW30XLE-440i. In two cases, 6.7 kWh/m^3^ was achieved, as the highest total energy value (see Table 8 and Table 11) and the lowest values was 0.7 kWh/m^3^.

It can also be observed that the yield increases with increasing feed temperature and permeate pressure. Although, the increase of feed temperature also poses extra energy demand. Increasing flow rate of recycled concentrate decreases yield for one module case. SW30XLE-440i has the highest yield, but it has the highest energy demand too, because the most materials are moved by this membrane.

It can be concluded that selecting the right membrane for appropriate desalination work is a difficult and complex task. Based on the present study, if the goal is to maximize the yield, SW30XLE-440i is recommended, while if the aim is to minimize the energy demand, the application of SW30XHR-440i membrane type is offered.

## 4. Conclusions

The software programs applied in the present study have not yet been compared in terms of desalination. The main advantages of these programs are the user-friendly panel manageability, furthermore that commercially available membranes can be modelled in the WAVE program, which sets it apart from other programs. To sum up, it can be deduced that the developed ChemCAD model and WAVE simulator are both suitable for desalination of 1000 m^3^/h initial process wastewater. The emission limit value, which is 0.05 V/V% NaCl, can be reached with optimized methods. The yield of MSF was 11.7% and in the case of RO it was between 3.0% and 12.6%.

The optimized multi-stage flash plant consists of three parts. 15 m^3^/h steam flow rate is required for optimal operation of the heat input section. The heat recovery section contains 10 flash stages and the heat rejection section includes three stages. It can be stated that the results connected to MSF can be generalized and can serve as a basis for later design of more complex and larger desalination plants. Three different reverse osmosis membrane modules were investigated: SW30XHR-440i, SW30HRLE-440i and SW30XLE-440i types. The choice between these membranes requires further consideration, since the membrane with the best yield also presented the highest energy demand. In this case, greater role is gaining ground in cost factors.

## Figures and Tables

**Figure 1 membranes-10-00265-f001:**
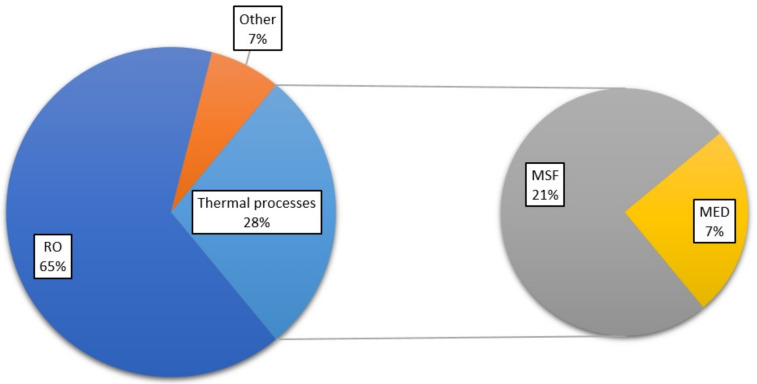
Distribution of desalination technologies.

**Figure 2 membranes-10-00265-f002:**
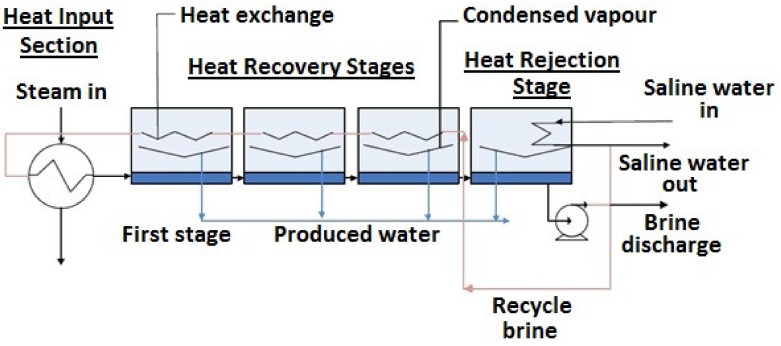
Schematic flowsheet of multi-stage flash distillation [11].

**Figure 3 membranes-10-00265-f003:**
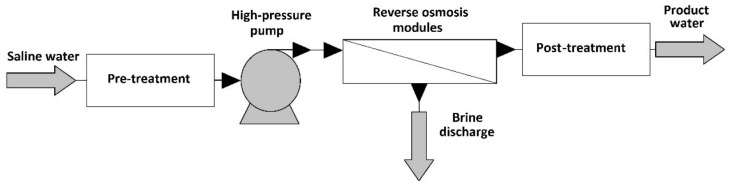
Schematic flowsheet of reverse osmosis plant [10].

**Figure 4 membranes-10-00265-f004:**
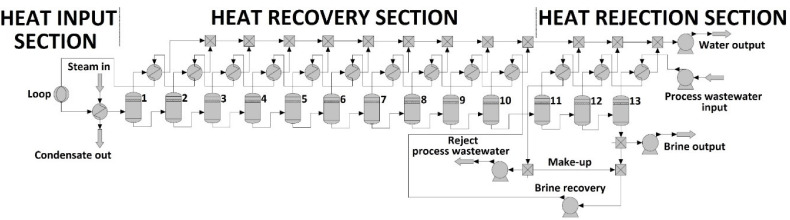
Flowsheet of multi-stage flash distillation method in ChemCAD simulator program.

**Figure 5 membranes-10-00265-f005:**
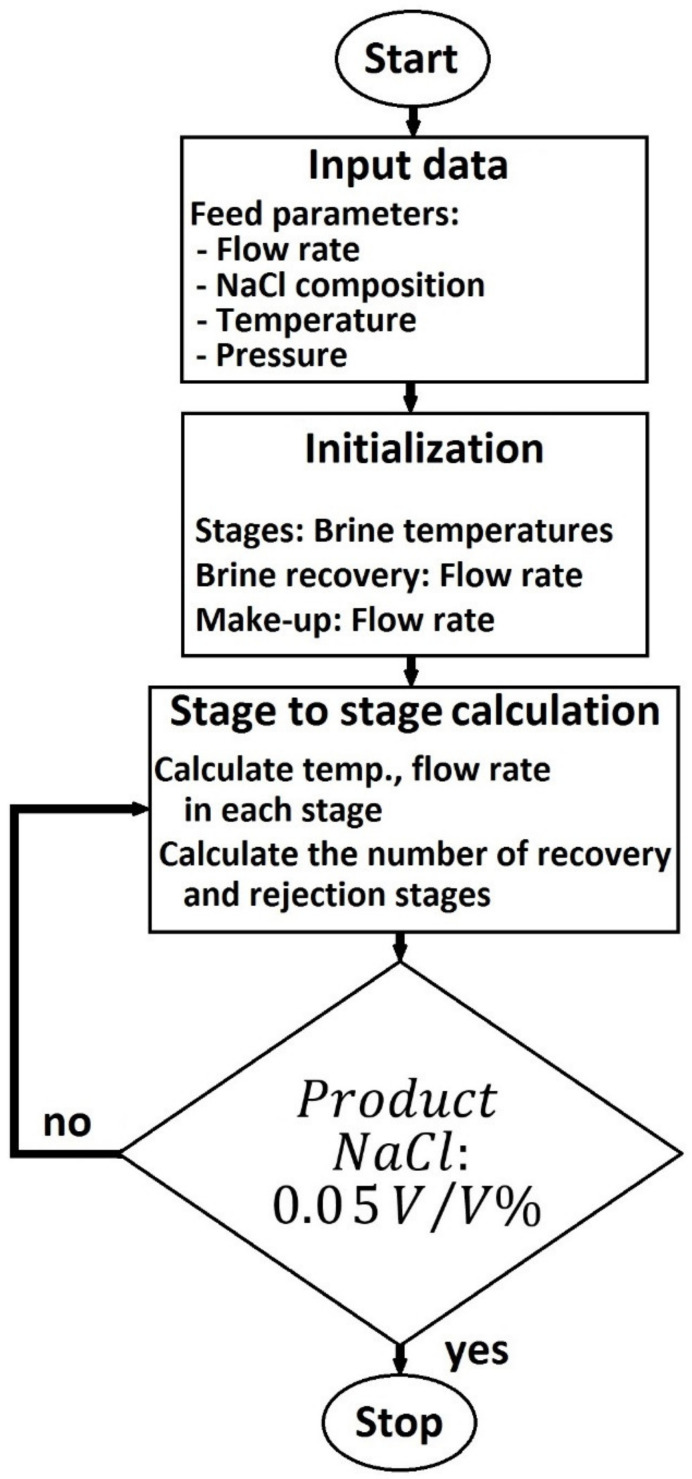
Flowchart of optimization process of multi-stage flash distillation.

**Figure 6 membranes-10-00265-f006:**
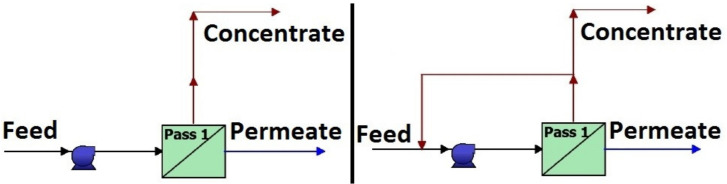
Flowsheets of reverse osmosis methods in WAVE program (right side: recycling version).

**Figure 7 membranes-10-00265-f007:**
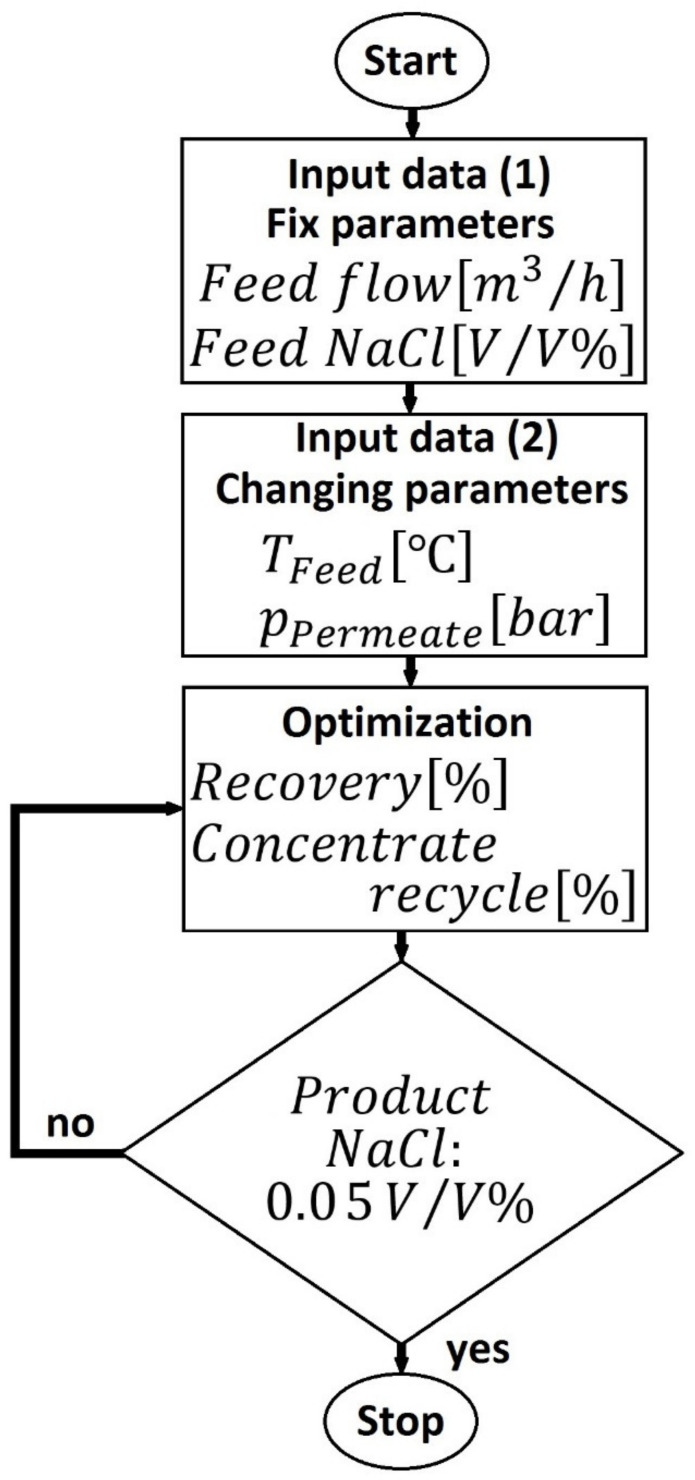
Flowchart of optimization process of reverse osmosis.

**Figure 8 membranes-10-00265-f008:**
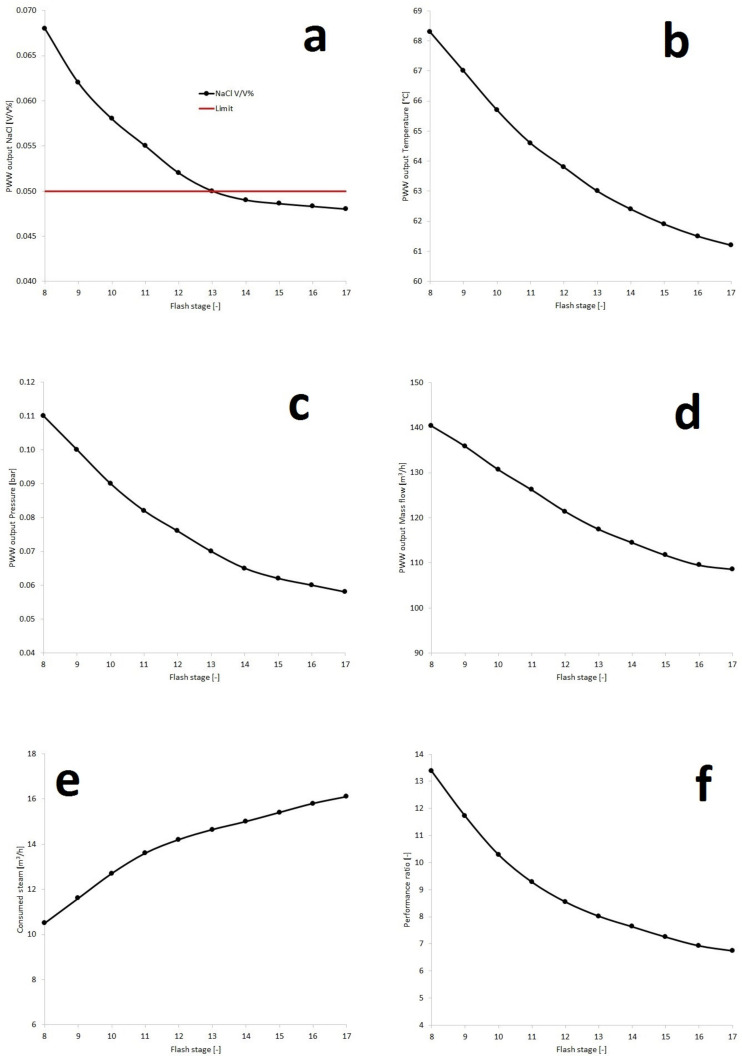
Influence of the number of stages on the water output NaCl (**a**), water output temperature (**b**), water output pressure (**c**), water output flow rate (**d**), consumed steam (**e**), performance ratio (**f**) and thermal energy (**g**).

**Figure 9 membranes-10-00265-f009:**
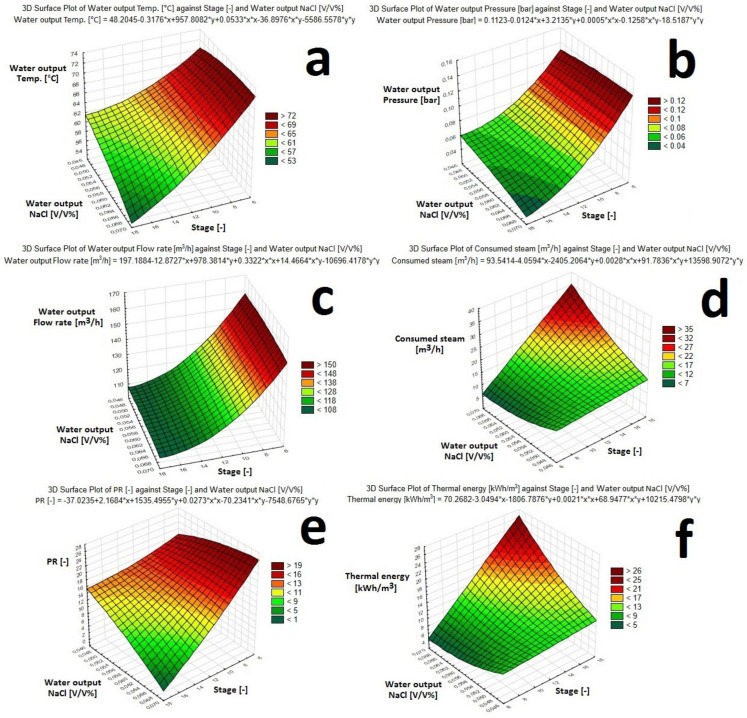
Influence of the number of Stages and Water output NaCl [V/V%] on the Water output Temperature (**a**), Water output Pressure (**b**), Water output Flow rate (**c**), Consumed steam (**d**), Performance ratio (**e**), and Thermal energy (**f**).

**Figure 10 membranes-10-00265-f010:**
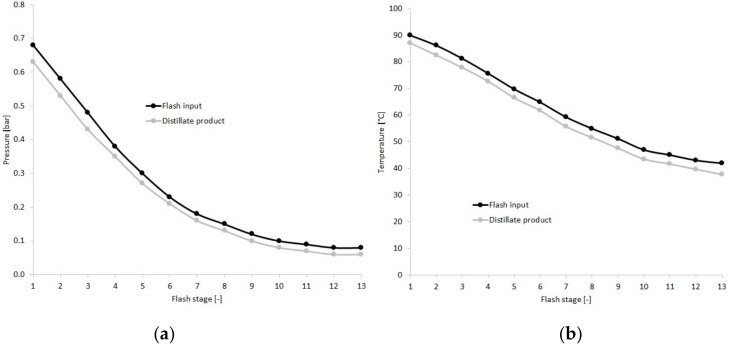
Flash input and distillation product parameters in the function of flash stage. (**a**): pressure, (**b**): temperature.

**Table 1 membranes-10-00265-t001:** Main features of desalination technologies [6].

Process	Thermal Energy [kWh/m^3^]	Electrical Energy [kWh/m^3^]	Total Energy [kWh/m^3^]	Investment Cost [USD/m^3^/d]	Total Water Cost [USD/m^3^]
**MSF**	7.5–12	2.5–4	10–16	1200–2500	0.8–1.5
**MED**	4–7	1.5–2	5.5–9	900–2500	0.7–1.2
**RO**	–	3–4	3–4	900–2500	0.5–1.2

**Table 2 membranes-10-00265-t002:** Productivity and unit water cost of some large SWRO plants [19].

SWRO Plant	Productivity [m^3^/day]	Unit Water Cost [USD/m^3^]
Ashkelon (Israel)	320,000	0.52
Palmachim (Israel)	83,000	0.78
Perth (Australia)	144,000	0.75
Carlsbad (California)	189,000	0.76
Skikda (Algeria)	100,000	0.73
Hamma (Algeria)	200,000	0.82
Hadera (Israel)	348,000	0.63

**Table 3 membranes-10-00265-t003:** Properties of examined reverse osmosis membrane modules.

Membrane Module	SW30XHR-440i	SW30HRLE-440i	SW30XLE-440i
Membrane type	Polyamide Thin-Film	Polyamide Thin-Film	Polyamide Thin-Film
	Composite	Composite	Composite
Active area [m^2^]	41	41	41
Max. operating pressure [bar]	83	83	83
Permeate flow rate [m^3^/d]	25	31	37.5
Min. salt rejection [%]	99.70	99.65	99.60
Stabilized salt rejection [%]	99.82	99.80	99.70
Max. operating temp. [°C]	45	45	45

**Table 4 membranes-10-00265-t004:** Comparison of AZ-ZOUR South MSF desalination plant results and ChemCAD calculations.

Parameters	Industrial Data [10]	Simulated Data	Error [%]
Top brine temperature [°C]	90.6	89.9	−0.7
Recycled brine flow rate [Ton/min]	238.10	238.12	0.01
Distillate produced [Ton/min]	18.80	18.82	0.09

**Table 5 membranes-10-00265-t005:** Optimized results of multi-stage flash distillation method.

Flow Rate	Temp. [°C]	Pressure [bar]	Flow Rate [m^3^/h]	Water [V/V%]	NaCl [V/V%]
PWW input	20.0	3.00	1000	95.50	4.50
Flash 1 input	89.9	0.70	1484	93.25	6.75
Flash 10 input	45.7	0.10	1378	92.67	7.26
Flash 13 output	40.0	0.07	1366	92.67	7.33
Make-up	36.5	2.50	304	95.49	4.51
Brine recovery	40.5	4.60	1484	93.26	6.75
Brine output	40.0	0.07	186	92.70	7.35
Reject PWW	36.5	0.07	696	95.51	4.51
Water output	63.0	0.07	117	99.95	0.05

**Table 6 membranes-10-00265-t006:** Optimized results of SW30XHR-440i membrane at 10 °C.

Feed Temp. [°C]	Concentrate Recycled [%]	Permeate Pressure [bar]	Total Energy [kWh/m^3^]	Yield [%]	Brine-Reduction [–]
10	0	10	0.7	4.3	0.99
10	0	20	0.8	4.4	0.99
10	0	30	1.0	4.5	0.99
10	30	10	4.4	3.0	0.99
10	30	20	5.2	3.2	0.99
10	30	30	5.9	3.3	0.99

**Table 7 membranes-10-00265-t007:** Optimized results of SW30XHR-440i membrane at 20 °C.

Feed Temp. [°C]	Concentrate Recycled [%]	Permeate Pressure [bar]	Total Energy [kWh/m^3^]	Yield [%]	Brine-Reduction [–]
20	0	10	0.9	6.4	0.99
20	0	20	1.0	6.5	0.99
20	0	30	1.2	6.6	0.99
20	60	10	4.5	5.2	0.99
20	60	20	5.0	5.3	0.99
20	60	30	5.6	5.4	0.99

**Table 8 membranes-10-00265-t008:** Optimized results of SW30XHR-440i membrane at 25 °C.

Feed temp. [°C]	Concentrate recycled [%]	Permeate pressure [bar]	Total Energy [kWh/m^3^]	Yield [%]	Brine- Reduction [–]
25	0	10	1.4	8.4	0.99
25	0	20	1.6	8.5	0.99
25	0	30	1.9	8.6	0.99
25	70	10	5.1	6.7	0.99
25	70	20	5.9	6.8	0.99
25	70	30	6.7	6.9	0.99

**Table 9 membranes-10-00265-t009:** Optimized results of SW30HRLE-440i membrane at 10°C.

Feed Temp. [°C]	Concentrate Recycled [%]	Permeate Pressure [bar]	Total Energy [kWh/m^3^]	Yield [%]	Brine- Reduction [–]
10	0	10	0.9	5.3	0.99
10	0	20	1.0	5.4	0.99
10	0	30	1.2	5.5	0.99
10	45	10	4.3	3.8	0.99
10	45	20	5.0	3.9	0.99
10	45	30	5.8	4.0	0.99

**Table 10 membranes-10-00265-t010:** Optimized results of SW30HRLE-440i membrane at 20 °C.

Feed Temp. [°C]	Concentrate Recycled [%]	Permeate Pressure [bar]	Total Energy [kWh/m^3^]	Yield [%]	Brine- Reduction [–]
20	0	10	1.1	8.3	0.99
20	0	20	1.3	8.4	0.99
20	0	30	1.4	8.6	0.99
20	70	10	5.0	6.7	0.99
20	70	20	5.7	6.8	0.99
20	70	30	6.5	6.9	0.99

**Table 11 membranes-10-00265-t011:** Optimized results of SW30HRLE-440i membrane at 25 °C.

Feed temp. [°C]	Concentrate Recycled [%]	Permeate Pressure [bar]	Total Energy [kWh/m^3^]	Yield [%]	Brine- Reduction [–]
25	0	10	2.0	10.4	0.99
25	0	20	2.4	10.5	0.99
25	0	30	2.8	10.6	0.99
25	75	10	5.2	7.9	0.99
25	75	20	6.0	8.0	0.99
25	75	30	6.7	8.1	0.99

**Table 12 membranes-10-00265-t012:** Optimized results of SW30XLE-440i membrane at 10 °C.

Feed Temp. [°C]	Concentrate Recycled [%]	Permeate Pressure [bar]	Total Energy [kWh/m^3^]	Yield [%]	Brine- Reduction [–]
10	0	10	1.1	6.2	0.99
10	0	20	1.3	6.4	0.99
10	0	30	1.5	6.5	0.99
10	55	10	4.1	4.6	0.99
10	55	20	4.9	4.7	0.99
10	55	30	5.7	4.8	0.99

**Table 13 membranes-10-00265-t013:** Optimized results of SW30XLE-440i membrane at 20 °C.

Feed Temp. [°C]	Concentrate Recycled [%]	Permeate Pressure [bar]	Total Energy [kWh/m^3^]	Yield [%]	Brine- Reduction [–]
20	0	10	1.4	10.4	0.99
20	0	20	1.6	10.5	0.99
20	0	30	1.9	10.5	0.99
20	75	10	4.6	7.9	0.99
20	75	20	5.4	8.0	0.99
20	75	30	6.2	8.1	0.99

**Table 14 membranes-10-00265-t014:** Optimized results of SW30XLE-440i membrane at 25 °C.

Feed Temp. [°C]	Concentrate Recycled [%]	Permeate Pressure [bar]	Total Energy [kWh/m^3^]	Yield [%]	Brine- Reduction [–]
25	0	10	2.0	12.4	0.99
25	0	20	2.4	12.5	0.99
25	0	30	2.8	12.6	0.99
25	80	10	5.0	11.4	0.99
25	80	20	5.8	11.6	0.99
25	80	30	6.6	11.8	0.99

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
