# Peer review of "Modelling and Optimisation of Multi-Stage Flash Distillation and Reverse Osmosis for Desalination of Saline Process Wastewater Sources"

_membranes, 2020, doi:10.3390/membranes10100265_

Round 1

Reviewer 1 Report

This paper provides an overview introduction of the three desalination technologies and compares the yields and energies required to reach the same product emission limit; The data and logic are clear. However, the paper is too detail in the introduction section, making it like a report or Wikipedia level writing. I don't think the authors need to focus on the detailed mechanism explanation of each method since these are common information, especially the principle of reverse osmosis. The authors should focus on the simulation and results. More data can be included, especially the verification of the optimized results from simulation with experimental data.

Author Response

Reviewer #1:

This paper provides an overview introduction of the three desalination technologies and compares the yields and energies required to reach the same product emission limit; The data and logic are clear. However, the paper is too detail in the introduction section, making it like a report or Wikipedia level writing. I don't think the authors need to focus on the detailed mechanism explanation of each method since these are common information, especially the principle of reverse osmosis. The authors should focus on the simulation and results. More data can be included, especially the verification of the optimized results from simulation with experimental data.

Answer:

I shortened the introduction part.

The MSF and RO calculations were considered as verified.

I added the following text to the manuscript:

“The developed MSF model was verified with industrial reference before calculation. The operating parameters of AZ-ZOUR South MSF desalination plant from the work of Al-Shayji [10] were examined.”

“WAVE 1.77a software was used for the calculation of reverse osmosis. The Water Application Value Engine (WAVE) is a modelling software program that integrates three of the leading technologies (ultrafiltration, reverse osmosis and ion exchange resin) into one comprehensive platform [38]. The calculation method of WAVE is not public. However, Altaee [32] presented in detail the calculation of its RO module (Reverse Osmosis System Analysis) with 95% accuracy, which is based on experimental and practical contexts. The equations are discussed in detail in the paper of Altaee [32]. Thus, program calculations can be considered verified. With the method, salt retention, yield, permeate concentration, and flow rate can be estimated per membrane module. The calculation is based on Van 't Hoff equation, average concentration factor, and it also takes into account the pressure drop of the concentrate side, salt diffusion coefficient and concentration polarization [32].”

“Table 4 shows the reference calculations of ChemCAD simulator. It can be seen that there is proper match between the results of industrial plant and ChemCAD simulations. Further parameters were also compared, which can be found in the Supplementary Part. Table S1 shows the temperature of recirculating brine entering each LLVF stage. Distillate produced from each stage in Ton/min can be found in Table S2. Finally, Table S3 shows the outlet pressure from each stage. There is minor difference between industrial and simulated data, in all cases. Thus, it can be established that the developed model is verified and capable of the calculation of MSF.”

Thank you for the appreciation of my work and thank you very much for your reviewer work.

Reviewer 2 Report

The submitted manuscript concerns an attractive subject. However, it presents several concerns limiting the quality of the work. Here are some points:

  • The abstract is general and lacks depth. More specific details on the work and its main findings should be given.
  • In the introduction some information are not concise, not updated or simply incorrect. For example,  MED unit capacity has reached the last years 15-20 MGD (67 000- 91000 m3/d). The information given in lines 47 to 51 should be checked and updated.
  • The details on RO (lines 97 to 159) are not relevant to the work. They should be shortened.
  • The literature review presents very few papers on MSF and RO modeling aspects with shallow and few relevant information. More papers should be included and their main findings are to be clearly exposed.
  • The research gaps are not identified and then the objectives of the work are vague and not clear.
  • The main contribution and novelty of the work are not identified.     

Author Response

Reviewer #2:

The submitted manuscript concerns an attractive subject. However, it presents several concerns limiting the quality of the work. Here are some points:

  1. The abstract is general and lacks depth. More specific details on the work and its main findings should be given.

Answer:

I revised the abstract part. More specific details (results and operating parameters, membrane types etc.) can be found in conclusions part too.

I added the following text to the manuscript:

“Nowadays, there is increasing interest in advanced simulation methods for desalination. The two most common desalination methods are multi-stage flash distillation (MSF) and reverse osmosis (RO). Numerous research works have been published on these separations, however their simulation appears to be difficult due to their complexity, therefore continuous improvement is required. The RO, in particular, is difficult to model, because the liquids to be separated also depend specifically on the membrane material. The aim of this study is to model steady-state desalination opportunities of saline process wastewater in flowsheet environment. Commercial flowsheet simulator programs were investigated: ChemCAD for thermal desalination and WAVE program for membrane separation. The calculation of the developed MSF model was verified based on industrial data. It can be stated that both simulators are capable of reducing saline content from 4.5 V/V% to 0.05 V/V%. The simulation results are in accordance with the expectations: MSF has higher yield, but reverse osmosis is simpler process with lower energy demand. The main additional value of the research lies in the comparison of desalination modelling in widely commercially available computer programs. Furthermore, complex functions are established between the optimized operating parameters of multi-stage flash distillation allowing to review trends in flash steps for complete desalination plants.”

“The software programs applied in the present study have not yet been compared in terms of desalination. To sum up, it can be deduced that the developed ChemCAD model and WAVE simulator are both suitable for desalination of 1000 m3/h initial process wastewater. The emission limit value, which is 0.05 V/V% NaCl, can be reached with optimized methods. The yield of MSF was 11.7% and in the case of RO it was between 3.0% and 12.6%.

The optimized multi-stage flash plant consists of three parts. 15 m3/h steam flow rate is required for optimal operation of the Heat Input Section. The Heat Recovery Section contains 10 flash stages and the Heat Rejection Section includes 3 stages. It can be stated that the results connected to MSF can be generalized and can serve as a basis for later design of more complex and larger desalination plants.

Three different reverse osmosis membrane modules were investigated: SW30XHR-440i, SW30HRLE-440i and SW30XLE-440i types. The choice between these membranes requires further consideration, since the membrane with the best yield also presented the highest energy demand. In this case, greater role is gaining ground in cost factors.”

  1. In the introduction some information are not concise, not updated or simply incorrect. For example, MED unit capacity has reached the last years 15-20 MGD (67 000- 91000 m3/d). The information given in lines 47 to 51 should be checked and updated.

Answer:

I added the following text to the manuscript:

“Although MED is thermodynamically more efficient than MSF, it is a less common method. MED units typically have a capacity of 600 to 91,000 m3/day, but involve high investment costs and energy consumption.”

I have checked the information in lines 47-51 and I have found these information correct. Table 1 is from a highly cited, serious source:

[6] N. Ghaffour, T.M. Missimer, G.L. Amy, Technical review and evaluation of the economics of water desalination: Current and future challenges for better water supply sustainability, Desalination, 309 (2013) 197-207.

  1. The details on RO (lines 97 to 159) are not relevant to the work. They should be shortened.

The literature review presents very few papers on MSF and RO modeling aspects with shallow and few relevant information. More papers should be included and their main findings are to be clearly exposed.

Answer:

I shortened the introduction part.

I have mentioned 12 following literature:

[21] F.E. Ahmed, R. Hashaikeh, A. Diabat, N. Hilal, Mathematical and optimization modelling in desalination: State-of-the-art and future direction, Desalination, 469 (2019) 114092.

[22] H.M. Ettouney, H. El-Dessouky, A simulator for thermal desalination processes, Desalination, 125 (1999) 277-291.

[23] W.L. Ang, A.W. Mohammad, 12 - Mathematical modeling of membrane operations for water treatment, in: A. Basile, A. Cassano, N.K. Rastogi (Eds.) Advances in Membrane Technologies for Water Treatment, Woodhead Publishing, Oxford, 2015, pp. 379-407.

[24] K.M. Sassi, I.M. Mujtaba, Effective design of reverse osmosis based desalination process considering wide range of salinity and seawater temperature, Desalination, 306 (2012) 8-16.

[25] M. Skiborowski, A. Mhamdi, K. Kraemer, W. Marquardt, Model-based structural optimization of seawater desalination plants, Desalination, 292 (2012) 30-44.

[26] H. Lv, Y. Wang, L. Wu, Y. Hu, Numerical simulation and optimization of the flash chamber for multi-stage flash seawater desalination, Desalination, 465 (2019) 69-78.

[27] G. Filippini, M.A. Al-Obaidi, F. Manenti, I.M. Mujtaba, Performance analysis of hybrid system of multi effect distillation and reverse osmosis for seawater desalination via modelling and simulation, Desalination, 448 (2018) 21-35.

[28] G.P. Rao, Unity of control and identification in multistage flash desalination processes, Desalination, 92 (1993) 103-124.

[29] A.M. Helal, M.S. Medani, M.A. Soliman, J.R. Flower, A tridiagonal matrix model for multistage flash desalination plants, Computers & Chemical Engineering, 10 (1986) 327-342.

[30] A. Husain, A. Woldai, A. Ai-Radif, A. Kesou, R. Borsani, H. Sultan, P.B. Deshpandey, Modelling and simulation of a multistage flash (MSF) desalination plant, Desalination, 97 (1994) 555-586.

[31] J. Belghaieb, W. Aboussaoud, M.-I. Abdo, N. Hajji, Simulation and Optimization of a Triple-Effect Distillation Unit, in, 2011.

[32] A. Altaee, Computational model for estimating reverse osmosis system design and performance: Part-one binary feed solution, Desalination, 291 (2012) 101-105.

I added the following text to the manuscript:

“It can be said that the above-mentioned specific models are able to reduce the salinity below the WHO desalination limit [33] for both membrane and thermal process cases. Not all simulator programs have built MSF calculation into their toolbars, which is also true for one of the most common chemical process simulators: ChemCAD does not have comprehensive desalination reference for complete plants. Reverse osmosis modelling is a membrane-specific process that requires complex computation, using calculations which are always valid for a given type of membrane.”

  1. The research gaps are not identified and then the objectives of the work are vague and not clear. The main contribution and novelty of the work are not identified.

Answer:

I added the following text to the manuscript:

“In conclusion, it can be observed that the simulation of MSF and RO is becoming more widespread. However, more commercial flowsheet simulator programs need to be investigated due to the increasing importance of desalination methods. The aim of this research is to investigate the desalination of saline process wastewater with multi-stage flash distillation in ChemCAD professional flowsheet simulator, and compare the results with those of reverse osmosis separation in WAVE design software from DuPont Company. Yields, operational properties, energetic factors, and complex functions between operating parameters were the basis for examination and comparison, which is considered as novelty in the case of mentioned computer programs.”

Reviewer 3 Report

It is a solid work offering a good overview of modelling of the specific separation techniques. However, I believe that a clear aim of this research should be formulated as well as the corresponding research question. In the actual form the presented manuscript is an exercise in modelling and optimization of rather complex task. So a scientific justification for this work should be added.

Author Response

Reviewer #3:

It is a solid work offering a good overview of modelling of the specific separation techniques. However, I believe that a clear aim of this research should be formulated as well as the corresponding research question. In the actual form the presented manuscript is an exercise in modelling and optimization of rather complex task. So a scientific justification for this work should be added.

Answer:

I added the following text to the manuscript:

“In conclusion, it can be observed that the simulation of MSF and RO is becoming more widespread. However, more commercial flowsheet simulator programs need to be investigated due to the increasing importance of desalination methods. The aim of this research is to investigate the desalination of saline process wastewater with multi-stage flash distillation in ChemCAD professional flowsheet simulator, and compare the results with those of reverse osmosis separation in WAVE design software from DuPont Company. Yields, operational properties, energetic factors, and complex functions between operating parameters were the basis for examination and comparison, which is considered as novelty in the case of mentioned computer programs.”

Thank you for the appreciation of my work and thank you very much for your reviewer work.

Reviewer 4 Report

This manuscript demonstrates the simulation of multi-stage flash distillation and reverse osmosis in process of saline water treatment by using the programs of ChemCAD and WAVE. The simulation results show a good agreement with those of corresponding industrial data. And also the comparison between the multi-stage flash distillation and reverse osmosis is conducted in terms of yield and energy consumption. The flowsheet simulator programs presented in this manuscript, to some extent, provides a new prospective to simulate and investigate the desalination of saline wastewater with multi-stage flash distillation reverse osmosis. But further revision is needed before acceptance. The author could consider the following comments.

  1. This is a research work, not a review. The introduction part is too long and needs to be further shortened. Like the introduction of RO principle is unnecessary, as it is well known to all.
  2. It is best to further state the importance or guiding significance of the simulation work in this manuscript. Please describe more detailedly the difference or advantage of the modeling method used, as compared to other reported methods.
  3. As the core of an article, the Results and Discussion has only about 600 words (excluding the Table and Figure legend), but the Sections of Introduction, Materials and Methods have more than 3000 words. This is not quite reasonable. It seems to lack corresponding discussion and analysis according to the results.
  4. In the main text, terms such as “multi-stage flash distillation”, “Reverse osmosis”… should be abbreviated as MSF, RO… because they have been already defined before.

Author Response

Reviewer #4:

This manuscript demonstrates the simulation of multi-stage flash distillation and reverse osmosis in process of saline water treatment by using the programs of ChemCAD and WAVE. The simulation results show a good agreement with those of corresponding industrial data. And also the comparison between the multi-stage flash distillation and reverse osmosis is conducted in terms of yield and energy consumption. The flowsheet simulator programs presented in this manuscript, to some extent, provides a new prospective to simulate and investigate the desalination of saline wastewater with multi-stage flash distillation reverse osmosis. But further revision is needed before acceptance. The author could consider the following comments.

  1. This is a research work, not a review. The introduction part is too long and needs to be further shortened. Like the introduction of RO principle is unnecessary, as it is well known to all.

Answer:

I shortened the introduction part.

  1. It is best to further state the importance or guiding significance of the simulation work in this manuscript. Please describe more detailedly the difference or advantage of the modeling method used, as compared to other reported methods.

Answer:

I added the following text to the manuscript:

“Not all simulator programs have built MSF calculation into their toolbars, which is also true for one of the most common chemical process simulators: ChemCAD does not have comprehensive desalination reference for complete plants. RO modelling is a membrane-specific process that requires complex computation, using calculations which are always valid for a given type of membrane.”

“The main advantages of these programs are the user-friendly panel manageability, furthermore that commercially available membranes can be modelled in the WAVE program, which sets it apart from other programs.“

  1. As the core of an article, the Results and Discussion has only about 600 words (excluding the Table and Figure legend), but the Sections of Introduction, Materials and Methods have more than 3000 words. This is not quite reasonable. It seems to lack corresponding discussion and analysis according to the results.

Answer:

I shortened the introduction part.

I revised the Results and Discussion part in the manuscript with discussion and analysis according to the results:

“Table 4 shows the reference calculations of ChemCAD simulator. It can be seen that there is proper match between the results of industrial plant and ChemCAD simulations. Further parameters were also compared, which can be found in the Supplementary Part. Table S1 shows the temperature of recirculating brine entering each LLVF stage. Distillate produced from each stage in Ton/min can be found in Table S2. Finally, Table S3 shows the outlet pressure from each stage. There is minor difference between industrial and simulated data, in all cases. Thus, it can be established that the developed model is verified and capable of the calculation of MSF.”

“Following the procedure of Figure 6, optimization of the MSF method was achieved. Seven different parameters were investigated in the function of flash stages:

  1. Water output NaCl [V/V%]
  2. Water output Temperature [°C]
  3. Water output Pressure [bar]
  4. Water output Flow rate [m3/h]
  5. Consumed steam [m3/h]
  6. Performance ratio: PR [-]
  7. Thermal energy [kWh/m3]

Figure 8 shows a two dimensional representation of tendencies. Figure 9 depicts complex functions between the mentioned parameters in three dimensional visualization, supplemented by the equation describing the plane.

It can be observed that the limit can be reached at the 13th flash stage, therefore the optimum point can be found here. The 11 kWh/m3 value in the case of the 13-step-plant fits into the literature tendency, as it can be seen in Table 1. NaCl composition, Temperature, Pressure, Flow rate and PR of Water output are almost on the same declining trend. The performance ratio was above 8 at the optimum location. In contrast, Consumed steam and required Thermal energy are following the opposite trend, which represent the cumulative heat effect of the system. These tendencies are in line with experience in the literature [18, 30]. The course of temperature and pressure are interrelated. In a binary system, one can be counted from the other. Furthermore, the values on the diagrams in Figure 8 can be traced back to the NaCl-Water phase diagram.

Table 5 introduces the optimized results of the MSF method. The steam flow was 15 m3/h. The brine reduction was 0.99 and 11.7% can be reached in yield value. Figure 10 shows the pressure and temperature values of flash inputs and distillate products of the stages between the 1st and the 13th. It can be seen that the curves are flattening close to the optimum region.”

“Table 6, Table 7 and Table 8 show the optimized results of SW30XHR-440i membrane. Table 9, Table 10 and Table 11 introduce the simulated data of SW30HRLE-440i membrane, and the results of SW30XLE-440i type RO membrane can be seen in Table 12, Table 13 and Table 14.

It can be affirmed that all three membranes met the limit value. As high as 0.99 brine reduction value can be reached in every case, which corresponds to 0.05 V/V% NaCl in permeate product. Studying the tables, low yield values are conspicuous. It means a characteristic result, which is the main disadvantage of RO compared to MSF. There is accordance between literature and simulated permeate flow rates or yields. SW30XHR-440i membrane has the lowest literature permeate flow rates and lowest simulated yields too. This is also the case for several other membranes, as it can be seen in Table 3.

The highest yield was obtained at the highest permeate pressure (30 bar) and 25°C. In the case of SW30XHR-440i membrane yield of 8.6% can be achieved. 10.6% was the maximum yield with SW30HRLE-440i membrane and 12.6% was in the case of SW30XLE-440i. In two cases, 6.7 kWh/m3 was achieved, as the highest total energy value (see Table 8 and Table 11) and the lowest values was 0.7 kWh/m3.

It can also be observed that the yield increases with increasing feed temperature and permeate pressure. Although, the increase of feed temperature also poses extra energy demand. Increasing flow rate of recycled concentrate decreases yield for one module case. SW30XLE-440i has the highest yield, but it has the highest energy demand too, because the most materials are moved by this membrane.

It can be concluded that selecting the right membrane for appropriate desalination work is a difficult and complex task. Based on the present study, if the goal is to maximize the yield, SW30XLE-440i is recommended, while if the aim is to minimize the energy demand, the application of SW30XHR-440i membrane type is offered.”

  1. In the main text, terms such as “multi-stage flash distillation”, “Reverse osmosis”… should be abbreviated as MSF, RO… because they have been already defined before.

Answer:

More “multi-stage flash distillation”, “Reverse osmosis” words were abbreviated.

Thank you for the appreciation of my work and thank you very much for your reviewer work.
